# A New Concept of Contiguous-Swath SAR Imaging with High Resolution: Strip-Spot SAR [note 1]

**DOI:** 10.3390/s22239153

**Published:** 2022-11-25

**Authors:** Furkan Korkmaz, Michail Antoniou

**Affiliations:** The School of Engineering, University of Birmingham, Birmingham B15 2TT, UK

**Keywords:** MIMO-SAR, high-resolution contiguous-swath, HRCS, Strip-Spot SAR

## Abstract

The study offers a Synthetic Aperture Radar (SAR) imaging concept called Strip-Spot SAR, which uses linear Multiple-Input Multiple-Output (MIMO) radar arrays, which are becoming increasingly attractive for short-range sensing in a variety of growing application sectors. The concept specifically employs Digital Beam-Forming (DBF) techniques, which are enabled in such systems to give contiguous azimuth imaging, as in Stripmap SAR, but with the fine spatial resolution of a Spotlight SAR. Its fundamental concepts are analytically derived and experimentally validated under laboratory conditions using calibrated and real targets. Finally, the performance and limitations of the concept are investigated.

## 1. Introduction

Depending on the user’s ultimate purpose, there are several SAR imaging modes available. Traditionally, the two major modes have been Stripmap and Spotlight SAR. In summary, a Spotlight SAR steers a high-gain antenna towards a target area during the SAR platform motion, which maximizes cross-range resolution and SAR sensitivity, as per Figure 1a. Conversely, a Stripmap SAR, as per Figure 1b, sweeps a relatively lower-gain antenna over the ground without beam steering and, as a result, suffers from a relatively lower sensitivity and coarser cross-range resolution, but with the advantage of being able to provide contiguous swath imagery [1].

A fundamental question that arises, then, is how both SAR spatial resolution and imaging swaths can be maximized while maintaining an acceptable sensitivity; this has recently been broadly coined as High-Resolution, Wide-Swath (HRWS) imaging and has been the subject of systematic research for the last few decades, e.g., [2,3]. To overcome this apparent contradiction, the multi-receive (sub-apertures) methods were first studied such as Digital Beamforming SAR (DBF-SAR) [4]. Then, the methods employing the MIMO array were considered such as MIMO-SAR, which gives multimodel mode operation including the aforementioned modes [5,6]. Different SAR modes, notably multi-channel Stripmap SAR, scan-SAR, and Terrain Observation with Progressive Scans SAR (TOPSAR), have been employed by using the Scan-On-Receive (SCORE) approach (primarily for space-borne applications). It has been proposed that illuminating a wider Doppler spectrum and dividing the receiver antenna into multiple azimuth apertures with independent receive channels can improve the azimuth resolution in multi-channel Stripmap mode. DBF may also be used in other areas, such as multi-aperture systems that employ the scan SAR or TOPSAR concepts to enable unambiguous wide swaths [7,8,9].

The new technology is linear MIMO arrays; these are used in cars, but can also be used in other mobile platforms, such as ground robots, drones, etc. It can be noted that most of the research that has been carried out offers ideas for the future, for instance parking lot detection [10,11,12,13], road border localisation [14], vegetation tracking along the roadside [15], road curvature [16], debris detection [17], and vehicle type categorization [12]. Additionally, the linear MIMO array is a low-cost system because of its low power consumption and small size, which increases the possibility of experimental studies. The following investigations could be interpreted as a system insight [18,19].

The aim of the above is to bring forward a MIMO SAR concept that allows mainstream, co-located, and linear MIMO radar arrays to maximize the SAR spatial resolution while maintaining contiguous swath widths. Specifically, the system resembles a Stripmap SAR on transmit by making use of individual low-gain transmit elements in the array; however, Spotlight SAR on receive performs DBF across the MIMO virtual array (Figure 1c). This allows multiple digital “Spotlight” beams to be formed, thereby enabling contiguous Spotlight SAR imaging across the path traversed by the SAR platform, referred to as “Strip-Spot” SAR.

In [20], the concept was first introduced in terms of system structure and a simulation based on a frequency of 24 GHz and point targets. In that study, the general approach to the concept was explained; however, the theoretical ground was only examined for a single target in a scene centre, and the algorithm obtained was only tested on calibrated targets. On the other hand, in this study, the algorithm were completed to cover every region of the target scene and was tested with real targets during the experimental stage. In addition, the performance values were examined as a complementary part of the concept.

The remainder of the paper is organised as follows: In Section 2, the whole Strip-Spot SAR concept will be explained, including the signal model and image formation sections. Then, Section 3 will give the results of the experiments at frequencies of 24 GHz and 77 GHz. Finally, performance characteristics such as sensitivity analysis, scene size, spatial resolution, and so on, will be investigated in Section 4.

## 2. Strip-Spot SAR

Figure 2 shows the layout of the system, which includes a number of scenes that are imaged at the same time and are close together. Each scene depicted here appears to correspond to the area shown during Spotlight imaging. Its value is proportional to the total distance/time travelled. On the other hand, the point towards which the radar is steered is referred to as the scene centre, $\left(X_{z},Y_{z}\right)$. The target position, $\left(X_{z}+X,Y_{z}+Y\right)$, can be set to any point in the scene.

The array has M transmitters and N receivers, with antenna element spacings of $d_{t}$ and $d_{r}$, respectively. Both can be set as λ/2, where λ is the operating radar wavelength, to maximize array gain and to avoid grating lobes; despite this, the MIMO array antenna element spacing condition allows one to create a fully filled virtual array from a fully filled and a sparse real array.

The instantaneous slant range, between the reference antenna array element (considered to be the first transmit element here) to a target in the *z*th scene and the target to the first receiver element, as per Figure 3a, can be calculated as:(1)$$R_{Rx}\approx R_{Tx}-d_{x}\sin\left(\theta \left(u,z\right)\right)$$
where *u* represents slow time, *z* represents the scene of interest, θ(u,z) denotes the steering angle at the radar position, at a time *u*, towards the *z*th scene, and $d_{x}$ is the distance between the first transmitter and first receiver antenna elements. This approximation is valid when $d_{x}<<R_{Tx}$.

Similarly, the simultaneous range between the *m*th transmitter and a target in the *z*th scene, as per Figure 3b, can be written as:(2)$$R_{Tx}^{\prime }\approx R_{Tx}-d_{m}\sin\left(\theta \left(u,z\right)\right)$$
where $d_{m}$ is the distance between first transmitter and *m*th transmitter and is equal to $\left(m-1\right)d_{t}$. Furthermore, the range for from the *z*th scene to the *n*th receiver, as per Figure 3c, can be written as:(3)$$R_{Tx}^{\prime }\approx R_{Tx}-d_{n}\sin\left(\theta \left(u,z\right)\right)$$
where $d_{n}$ is the distance between the first transmitter and *n*th receiver and is equal to $\left(n-1\right)d_{r}$. The slant range between every other array element and the same target location as the platform moves is given by $R_{mn}\left(u,z\right)$, such that:(4)$$\begin{array}{cc} R_{mn}\left(u,z\right) & \approx R_{Tx}^{\prime }+R_{Tx}^{\prime }\\ \; & \approx R_{Tx}-d_{m}\sin\left(\theta \left(u,z\right)\right)+R_{Tx}-d_{n}\sin\left(\theta \left(u,z\right)\right)\\ \; & =R_{Tx}-d_{m}\sin\left(\theta \left(u,z\right)\right)+R_{Tx}-d_{x}\sin\left(\theta \left(u,z\right)\right)-d_{n}\sin\left(\theta \left(u,z\right)\right)\\ \; & =2R_{Tx}-\left(d_{m}+d_{x}+d_{n}\right)\sin\left(\theta \left(u,z\right)\right) \end{array}$$

To simplify Equation (Equation 4), it can be combined as:(5)$$d_{mn}=d_{m}+d_{x}+d_{n}$$

Therefore, the final version of the slant range equation, referenced to the first transmitter, can be written as:(6)$$R_{mn}\left(u,z\right)\approx 2R_{Tx}-d_{mn}\sin\left(\theta \left(u,z\right)\right)$$

Considering that commercial MIMO radar arrays operate at frequencies in the order of tens of gigahertz (typically 24 or 77 GHz) and typical stand-off distances are in the region of a few hundred of metres, this approximation can be safely made.

### 2.1. Signal Model

In line with the specifications of current short-range MIMO arrays (e.g., those commonly used in the automotive environment [14,21,22]), it is assumed the transmitted waveform has a linear frequency modulation and that the overall MIMO transmission scheme is Time Division Multiple Access (TDMA). A simplified expression for the signal transmitted from the $m^{\mathrm{th}}$ transmitter can thus be written as:(7)$$s_{m}\left(t\right)=w\left(t\right)\cos\left(2\pi f_{c}t+\pi Kt^{2}\right)$$
where w(t) is the signal envelope, $f_{c}$ is the radar carrier frequency, *t* is fast time, and *K* is the chirp (or sweep) rate of the signal. The signal echo from each transmit element, received at each receive element, and the output of the $n^{\mathrm{th}}$ receive element are given by:(8)$$s_{mn}\left(t,u,z\right)=w\left(t-t_{mn}\left(u,z\right)\right)w_{a}\left(u-u_{c}\right)\cos(2\pi f_{c}\left(t-t_{mn}\left(u,z\right)\right)+\pi K_{r}\left(t-t_{mn}{\left(u,z\right)}^{2}\right)$$
where $s_{mn}\left(t,u,z\right)$ is the SAR signal received from a transmit–receive element pair, *m* and *n* are the transmitter and receiver indices, respectively, $u_{c}$ is the time of zero Doppler crossing, $w_{a}$ is the envelope of the signal in the azimuthal direction, and $td_{mn}\left(u,z\right)$ is the time delay for a single target between each transmitter and receiver pair in each scene and varies with slow time as:(9)$$td_{mn}\left(u,z\right)=\frac{R_{mn}\left(u,z\right)}{c}$$
where $R_{mn}\left(u,z\right)$ is the instantaneous slant range between a given transmit–receive element pair and a target at scene *z*, and *c* is the speed of light. It can be seen that the slant range is a function of *u* and *z*. As an example, for a target at the $z^{\mathrm{th}}$ scene, with range and cross-range co-ordinates of $\left(Y_{z}+Y,X_{z}+X\right)$, respectively, $R_{mn}$ can be calculated by:(10)$$R_{mn}\left(u,z\right)=2\sqrt{{\left(Y_{z}+Y\right)}^{2}+{\left(vu-\left(X_{z}+X\right)\right)}^{2}}-d_{mn}\sin\left(\theta \left(u,z\right)\right)$$
where *v* is the platform velocity. The term vu is the distance of the platform taken in the cross-range and is with respect to the reference physical array element. Using a Taylor approximation in Equation (Equation 10) for the first term, it can be written as:(11)$$R_{mn}\left(u,z\right)\approx 2\left(Y_{z}+Y\right)+\frac{{\left(vu-\left(X_{z}+X\right)\right)}^{2}}{\left(Y_{z}+Y\right)}-d_{mn}\sin\left(\theta \left(u,z\right)\right)$$

Substituting Equation (Equation 11) into Equation (Equation 8) then yields:(12)$$\begin{array}{c} s_{mn}\left(t,u,z\right)=w\left(t-td\left(u,z\right)\right)w_{a}\left(u-u_{c}\right)\cos\left(-j\frac{4\pi (Y_{z}+Y)}{\lambda }\right)\cos\left(-j\pi \frac{{\left(vu-\left(X_{z}+X\right)\right)}^{2}}{\lambda (Y_{z}+Y)}\right)\\ \cos\left(j\frac{2\pi }{\lambda }d_{mn}\sin\left(\theta \left(u,z\right)\right)\right)\cos\left(j\pi K_{r}{\left(t-td_{mn}\left(u,z\right)\right)}^{2}\right) \end{array}$$

The second term in Equation (Equation 10) is much smaller than the first, which shows that the latter is negligible in terms of time, but not in terms of phase. Hence, time delay in the envelope is approximated as being equal to the first term in Equation (Equation 10). After quadrature demodulation (or an additional Hilbert transform if data are recorded at Intermediate Frequency (IF)), Equation (Equation 12) can be rewritten as:(13)$$\begin{array}{c} s_{mn}\left(t,u,z\right)=w\left(t-td\left(u,z\right)\right)w_{a}\left(u-u_{c}\right)\exp\left(-j\frac{4\pi (Y_{z}+Y)}{\lambda }\right)\exp\left(-j\pi \frac{{\left(vu-\left(X_{z}+X\right)\right)}^{2}}{\lambda (Y_{z}+Y)}\right)\\ \exp\left(j\frac{2\pi }{\lambda }d_{mn}\sin\left(\theta \left(u,z\right)\right)\right)\exp\left(j\pi K_{r}{\left(t-td_{mn}\left(u,z\right)\right)}^{2}\right) \end{array}$$

In Equation (Equation 13), the first exponential term is a constant, the second term contains the azimuth phase history of the SAR signal, and the third is used for beamforming in the azimuth. The digital beamforming term in the equation appears due to the co-located MIMO array, which gives rise to the difference with a conventional SAR signal. It is this term that is ultimately exploited to focus beams simultaneously pointed towards each scene, *z*, as the platform moves in slow time, *u*. A broadside model is assumed here for simplicity; however, the use of broad-beam antennas for signal transmission, with digital beamforming on receive, allows squint mode acquisitions to also be performed.

### 2.2. Image Formation

The stages of the image formation algorithm are shown in Figure 4 and consist of the digital beamforming, range compression, interpolation, and backprojection steps. This process is repeated for each scene.

Azimuth DBF can be used on the SAR raw data (Equation (Equation 13)) to create a narrow and steerable beam towards the target of interest whilst maximizing the Signal-to-Noise Ratio (SNR) and spatial resolution [4]. The steering vector [23] can be written as:(14)$$AF_{mn}\left(u,z\right)=\exp\left(-j\frac{2\pi }{\lambda }d_{mn}\sin\left(\theta \left(u,z\right)\right)\right)$$

Equation (Equation 14) shows that a beam can be steered towards the centre of the $z^{th}$ scene as the SAR platform advances in *u*, such that acquisition is similar to that of Spotlight SAR. More importantly, Equation (Equation 13) shows that the azimuthal beamforming term varies with the illuminated scene *z* as θ(u,z) varies, and Equation (Equation 14) can be used to simultaneously form multiple such beams for each scene, thus resulting in multiple Spotlight SAR acquisitions on adjacent scenes and, hence, creating contiguous SAR images with a Spotlight SAR resolution. The DBF operation on a single scene at a single position along the synthetic aperture can be written as:(15)$$s\left(t,u,z\right)=\sum_{m=1}^{M}\sum_{n=1}^{N}s_{mn}\left(t,u,z\right)\left(AF_{mn}\left(u,z\right)\right)$$
which results in:(16)$$\begin{array}{c} s\left(t,u,z\right)=MNw\left(t-td\left(u,z\right)\right)w_{a}\left(u-u_{c}\right)\exp\left(-j\frac{4\pi (Y_{z}+Y)}{\lambda }\right)\\ \exp\left(-j\pi \frac{{\left(vu-\left(X_{z}+X\right)\right)}^{2}}{\lambda (Y_{z}+Y)}\right)\exp\left(j\pi K_{r}{\left(t-td\left(u,z\right)\right)}^{2}\right) \end{array}$$

Equation (Equation 16) now resembles the signal received in Spotlight SAR, and thus, every scene *z* can be processed with any standard image formation algorithm. Since this will be detailed in Section 4 in terms of the sensitivity characteristics, only MN was used as a factor. Here, the standard backprojection was followed; however, any Spotlight SAR algorithm could be used [24,25]. The first step is range compression, which can be implemented by matched filtering in the fast-time frequency domain:(17)$$s_{rc}\left(f,u,z\right)=FFT\left(s\left(t,u,z\right)\right)*conj\left(FFT\left(s_{m}\left(t\right)\right)\right)$$
where $s_{rc}\left(f,u,z\right)$ is the range-compressed signal in the frequency domain. This yields:(18)$$\begin{array}{c} s_{rc}\left(t,u,z\right)=MNp_{r}\left(t-td\left(u,z\right)\right)w_{a}\left(u-u_{c}\right)\exp\left(-j\frac{4\pi (Y_{z}+Y)}{\lambda }\right)\\ \exp(-j\pi \frac{{\left(vu-\left(X_{z}+X\right)\right)}^{2}}{\lambda (Y_{z}+Y)}) \end{array}$$
where $p_{r}$ is the range-compressed signal envelope. Prior to backprojection, interpolation is applied in the range direction. Following a pixel-by-pixel approach, the image reflectivity at the target co-ordinates can be expressed by:(19)$$f\left(X_{z}+X,Y_{z}+Y\right)=\sum_{u}s_{rc}\left(td\left(u,z\right),u,z\right).$$

Equation (Equation 19) gives an image for a specific scene. Therefore, the backprojection process can be repeated to obtain the images from the adjacent scenes. The distance migration caused by the cluster radar’s stripe processing is also compensated by the backprojection algorithm. The analysis does not include motion errors; however, even in the presence of such, it is expected that their compensation will be similar to, if not the same as in standard Spotlight SAR.

## 3. Experiments

In this section, multiple experiments are conducted to verify the Strip-Spot concept. So that the results could be compared, simulations were performed that mimicked the experimental setup. Before proceeding with the experiments, information on the equipment utilised, such as radar and targets, is provided. The experiments are primarily divided into two sections. The experiment with corner reflectors run at 24 GHz will be presented first, followed by an experiment with an extended target, performed at 77 GHz. The second experiment is also divided into two parts: the target of the bike and the bike with the corner reflector.

The experiments used a commercial, off-the-shelf linear MIMO array radar operating at 24 AND 77 GHz with two different front-ends, as per Figure 5. The first of these has two transmitters and eight receivers. The latter has four transmitters and eight receivers. It performs according to the parameters given in Table 1.

### 3.1. Experiment at 24 GHz

A series of experiments was conducted in the laboratory environment to verify the Strip-Spot concept. To begin, the first experiment was conducted at a centre frequency of 24 GHz. Figure 6 depicts the experimental setup. It can be seen that three targets are located in three distinct settings, all at 0.5 m intervals, at a minimum distance of 5 m apart. The cross-range reference point was the centre target position, and the total aperture length employed was 2 m.

The front-end used in the experiment has for each element a very wide beamwidth in the azimuth of 76.5° and a narrow beam of elevation (12.8°). The total length of the array was approximately 70 mm, resulting in an array azimuth beamwidth of approximately 10°.

The radar was mounted on a linear positioner with an active length of 2 m to provide platform motion as in Figure 7. Due to the high frequency and the expected fine spatial resolution, the positioner was electronically controlled with a horizontal accuracy of 1 mm. Measurements at each aperture position were performed in a stop-and-go fashion.

With a total array beamwidth of roughly 10° and a distance of 5 m, a conventional Spotlight SAR would be physically unable to image all three targets simultaneously due to beam pointing, whereas this should be possible with the proposed concept (Figure 8a). In line with the theoretical background, it can thus be envisaged that three scenes are simultaneously imaged, and each corner reflector was at the centre of each scene. Hence, the first and primary step was to process the recorded data with the proposed Strip-Spot algorithm.

Subsequently, a second experiment was conducted for comparison. In this, only a single transmit element was used and a receive broadside beam was formed from the eight receive elements (Figure 8b), which resembled a Stripmap SAR acquisition and could hence be processed in the according manner. This allowed comparing the proposed imaging mode to a conventional SAR imaging mode.

#### Results

The experimental image obtained by applying the Strip-Spot algorithm to the experimental data is shown in Figure 9a.

For comparison, the simulated results, as obtained by applying the same algorithm to the simulated data with the same experimental parameters, are shown in Figure 9b. The colour scale in both images is in dB, with 0 dB representing the highest compressed echo intensity in each image.

The images in Figure 9 show a high similarity between the simulated and experimental results, and all targets appear at their expected locations. In the experimental image, it can be seen that all three targets can be observed, which verifies the expectation of the algorithm in terms of its ability to provide contiguous SAR imagery. It can also be seen that the point spread functions obtained from the leftmost and rightmost targets appear squinted, while the middle target does not. This, too, is expected, since with the experimental setup available, the range histories of both targets are asymmetric (Figure 8). Furthermore, the echoes visible in the image’s far range derive from the wall. They are reduced behind the interval from the leftmost to the centre corner reflector, which is due to the presence of a carpet, which can be seen in Figure 6.

The next key item to verify the Strip-Spot concept is the resolution obtained. Figure 10 shows the cross-sections of the images in Figure 9 in the cross-range direction, at a distance of 5 m. From the response of the middle target, a cross-range resolution of approximately 1.58 cm could be measured. For Spotlight SAR, the cross-range resolution is given by:(20)$$d_{Spotlight}=\frac{\lambda R}{2L}=1.56\mathrm{cm}$$
where *R* is the target distance of 5 m and *L* is the total aperture length of 2 m.

The obtained Strip-Spot resolution is thus in good accordance with that of a Spotlight SAR, as expected.

As a sanity check, for the Stripmap acquisition conducted in our second experiment, the cross-range resolution can be calculated to be half the length of the receive MIMO array, i.e., approximately 2.88 cm, which is substantially different from that measured here. Figure 11 depicts the image under the Stripmap processing for both the simulation environment and experiment. Similarly, all targets appear at the expected locations.

On the other hand, the cross-range resolution is expected to be 2.88 cm, as mentioned above. To confirm this result, Figure 12 gives the cross-sections of the images for the simulated and experimental results. In the experiment, the cross-range resolution was measured to be 2.90 cm. It can, therefore, be stated that the Strip-Spot resolution is that of a Spotlight, rather than Stripmap SAR, and under the specific experimental conditions, the improvement in cross-range resolution compared to a Stripmap SAR is approximately 91%. The improvement comes from the Spotlight processing, which evaluates the data taken by longer aperture lengths. In other words, the integration time affecting the cross-gap resolution was almost doubled compared to the Stripmap mode.

## 4. Experiments at 77 GHz

After completing first the verification experiments with calibrated targets at 24 GHz, the concept’s response for a higher frequency and under more realistic scenarios was considered. The expecting cross-range resolution would be tripled according to the images at 24 GHz.

In this experiment, a bicycle was utilised as the real-life target. Figure 13 illustrates the experimental design for the investigation. As may be observed, a total aperture length of 6 m was employed. The target bike was vertically symmetrically positioned relevant to the pedals of the target bike. The bike measures 1.6 m × 1.1 m in size. Essentially, two scenes were employed, with the scene centres at (0.4,3.8) and (−0.4,3.8) serving as the steering points for the beams.

The bicycle, which is our target in this experiment, as shown in Figure 14, was placed in the experiment area at an angle symmetric with respect to the total length. The distance to the closest point of the bike to the linear positioner is 3.2 m.

The radar with a centre frequency of 77 GHz and a bandwidth of 2 GHz was used in this experiment. It was difficult to determine the resolution due to the high operating frequency of the radar used and the effect created by the near-field. For this purpose, a corner reflector was placed in front of the bike to allow for a comparison assessment.

As for how the aperture length was determined, this was carried out according to the Strip-Spot concept, on the physical basis of Spotlight imaging. More clearly, the steering angle was chosen to scan from about −45° to 45°. Since the minimum range is 3.2 m in this instance, approximately 6 m was considered sufficient.

### 4.1. Results

The first experiment’s results are depicted in Figure 15. Figure 15a shows the result of the Stripmap mode imaging via the broadside DBF beam, whilst Figure 15b shows the results of the Strip-Spot concept. The target can be seen clearly and in considerable detail in Strip-Spot mode. For example, while the D lock on the bike cannot be clearly seen in the first image, the second image shows the bottom corner of the lock.

When Figure 15b and the bicycle photo in Figure 14b, as taken prior to the experiment, are superimposed, it can be seen that the picture and the photo overlap in Figure 16. It can also be seen that different reflections are gained from different materials. For example, the reflectivity of the metal parts of the bicycle, which itself consists of metal, aluminium, plastic, and rubber, is higher, which is the expected result.

Although the target is clearly displayed, it is very difficult to verify cross-range resolution. As stated in the previous section, a second experiment was performed on the same scene, the only difference being that a high-reflectivity corner reflector was placed just to the right of the bicycle pedal. The results of this experiment can be seen in Figure 17. As in Figure 15, the two results are evaluated as “Stripmap” and “Strip-Spot.” Although the corner reflector dynamic range also changes, the bike is seen to the same extent.

When one looks at the corner reflector in Figure 17a, it seems that some part of it is lost due to interference with the top tube of the bike. When looking specifically at the corner reflector, it can be observed that the sidelobes are evenly distributed in both the range and the cross-range. When looking at the Strip-Spot image in Figure 17b, both the corner reflector and the bike can be clearly seen, and it was observed that the interference at the top tube was not present in this situation.

Figure 18, which shows the cross-section of the corner reflector from the first experiment, can be used to evaluate the concept’s cross-range resolution.

The cross-range resolution for the corner reflector in Figure 18, on the other hand, is 6.97 mm for the Stripmap and 3.53 mm for Strip-Spot concept.

### 4.2. Discussion

When the results obtained in the various experiments are brought together, Table 2 can be constructed. The Strip-Spot concept promises to be able to obtain images on the low millimetre scale.

According to the results of two different frequency trials, the cross-range resolutions for both modes, Stripmap and Strip-Spot, at 24 GHz, are nearly identical. In either case, the outcome of processing in only the Stripmap mode nearly matches the expected cross-range resolution in the 77 GHz experiment. This was not the case, however, with the Strip-Spot concept. Again, the finer resolution is approximately double the cross-range resolution of the Stripmap.

There are effects of the near-field and very fine resolution, so the experimental cross-range resolution is difficult to theoretically calculate. However, it is expected that, as in the 24 GHz experiments, the Strip-Spot resolution should be roughly double that of Stripmap. The experimental results appear to confirm this expectation.

## 5. Performance Analysis

Every system or model has inherent limitations. Of course, this reality also applies to the Strip-Spot concept. To ascertain its limits, performance parameters such as sensitivity analysis and image resolutions can be examined. Beginning with a sensitivity analysis, which will be derived for Strip-Spot-SAR, this section will attempt to determine the maximum angle at which the steering can operate without sacrificing performance, which is accomplished through digital beamforming. Then, the optimal scene size will be investigated.

### 5.1. Sensitivity Analysis

The Noise-Equivalent Sigma Zero (NESZ) sensitivity equation for SAR systems [26] can be modified to account for the MIMO array configuration as follows:(21)$$\sigma^{0}\left(NESZ\right)=\frac{2V{\left(4\pi \right)}^{3}R^{3}kTFL_{s}}{P_{av}G_{Tx}G_{Rx}\lambda^{3}d_{r}}\frac{1}{MN}.$$
where *V* is the platform velocity, *R* is the range, *k* is the Boltzmann constant, *T* is thermal noise, *F* is the noise figure, $L_{s}$ is the signal loss, $P_{av}$ is the average power, $G_{Tx}$ and $G_{Rx}$ are the transmit and receive antenna gain, and *M* and *N* are the number of transmitters and receivers, respectively. Due to the array structure, the gains for the transmitting and receiving antenna elements are multiplied by the number of transmitter and receiver elements.

#### 5.1.1. Sensitivity of the Strip-Spot Concept

Following the determination of the sensitivity equation for MIMO-SAR, the current sensitivity state for the Strip-Spot design is examined. While transmission is carried out in Stripmap mode, reception is carried out in Spotlight mode. Consequently, the receive antenna array is directing via the motion of the SAR platform to point toward the centre of a target region. The steering creates a difference in the sensitivity measurement of the instrument. As a result, there is a scan loss that is variable in nature.

Antenna gain was examined in order to find the scan loss caused by this steering. Antenna gain is dependent on the frequency and the effective area of the antenna. In addition, the effective antenna area is defined by the beamwidth in both the elevation and azimuth axes. As a result, changes in beamwidth have an explicit influence on antenna gain.

If there is an array, the gain is affected via containing the effect of the beamwidth above, so array gain can be calculated by:(22)$$G_{a}=G_{e}\times AF$$
where $G_{a}$ represents the total array gain, $G_{e}$ represents antenna element gain, and AF represents the array factor.

For SAR consideration, each sampling point is a different steering angle at the same time, so the beamwidth is expected to be different for each sampling point. Finally, the receive array gain does not remain identical for each aperture position.

DBF is applied to the receiver array. In terms of the array factor, the value is dependent on the steering angle, which is determined by the position of the radar platform. When there is no steering, the value is maximised. No steering means that the beam is aimed at the broadside. As a result, the maximum gain is the product of the transmitter and receiver gains multiplied by the element numbers for the transmitter, *M*, and receiver, *N*. When steering is used, antenna gain is expected to be reduced. Each transmitter fulfils transmission in this case, and there is no beamforming. The array factor [27] can be written for our scenario:(23)$$AF\left(\theta ,\theta_{uz}\right)=\frac{\sin\left(\frac{N\pi (sin\left(\theta \right)-sin\left(\theta_{uz}\right)}{2}\right)}{N\sin\left(\frac{\pi (sin\left(\theta \right)-sin\left(\theta_{uz}\right)}{2}\right)}$$
where *z* is the scene number, *u* is the azimuth sampling number, and θ is the steering angle. There is an array factor for each azimuth position that corresponds to *u*. The total array factor can be updated to:(24)$$AF=\sum_{u=1}^{U}AF\left[\theta ,\theta_{uz}\right]$$
where *u* and *U* represent the aperture position and, then, the total number of aperture positions and $AF[\theta ,\theta_{uz}]$ represents the array factor at a position *u* for scene *z*. The steering angle varies depending on the aperture and scene position.

The initial grazing angle, θ, is typically believed to be $90^{\circ }$ broadside. According to the best-case scenario, the steering angle, $\theta_{uz}$, is also zero degrees, which represents no steering. In summary, when there is no steering, the gain for the transmit array takes a maximum value, which may be expressed as $G_{TM}$. Total receiver antenna gain, on the other hand, is affected by the array factor. The normalized and averaged array factor, $AF_{av}$, is assigned a number between 0 and 1. As a result, the maximum gain may be calculated as the element receiver’s gain multiplied by *N*, and so, the minimum gain will be zero.
(25)$$G_{Total}=G_{Rx}\left(NAF_{av}\right)$$
where $G_{Total}$ denotes the receiver array antenna gain, $G_{Rx}$ denotes the single receiver antenna gain, and $AF_{av}$ denotes the array gain factor.

If Equation (Equation 25) is substituted into Equation (Equation 21), the final sensitivity equation can be written as:(26)$$\sigma^{0}\left(NESZ\right)=\frac{2V{\left(4\pi \right)}^{3}R^{3}kTFL_{s}}{P_{av}G_{Tx}MG_{Rx}\left(NAF_{av}\right)\lambda^{3}d_{r}}.$$

To extend this, the sensitivity for a single Tx/single Rx (Stripmap), single Tx/Rx array (Stripmap), Spotlight (Tx/Rx on both arrays), and Strip-Spot can be seen in Table 3. In this manner, one can see where the Strip-Spot concept stands as a performance parameter by comparing the Stripmap with the Spotlight modes. The results of this comparison can be seen in the following section.

#### 5.1.2. The Case Study for the Experiment

The sensitivity of the experiment may be shown using the experimental parameters, as shown in Table 4. As previously mentioned, the radar features four transmitter and eight receiver antennas that operate at a frequency of 77 GHz. When performing the experiment, a 2 GHz bandwidth was used.

Figure 19 depicts the outcomes of the four options considered. As it turns out, the single Tx/single Rx Stripmap sensitivity is by far the poorest of any of the options considered. This receiver portion can be improved by roughly 9 dB when it is equipped with an array of components. It is comparable to the Spotlight option in terms of the sensitivity value. The crucial and promising finding is that the Strip-Spot concept is only 1 dB behind the leading mode.

The key here is that Strip-Spot has a sensitivity close to, but slightly worse than, Spotlight. Furthermore, assuming the acceptable SAR sensitivity to be −25 dB, this gives us an acceptable operating range for this type of radar and this type of imaging mode to about 190 m. This range can be considered ideal for short-range applications. It gives an indication of the feasibility of the concept. Furthermore, this result confirms that the system promises high-resolution continuous-swath imaging up to the range above, in the range of $\pm 45^{\circ }$.

### 5.2. Scene Size

Consider the case when a single transmit element generates a broad beam. DBF was employed on the receiving side of that wide transmit beam. As a consequence, the entirety of the illuminated area is split into discrete scenes with our receive array beam oriented at each aperture location, and the beam broadens to a variable extent. To find out if there is a significant effect of widening scene size on the concept, the size of specific scenes will be considered in this section. Initially, the scene size can be found as:(27)$$SS\approx R\Theta_{e}\times R\Theta_{as}$$
where SS is the scene size, *R* is the range, and $\Theta_{e}$ and $\Theta_{as}$ are the beamwidth in the elevation and azimuth, respectively.

However, it is different for the target of interest if performing Spotlighting or there is a look angle. Hence, the broadening in the azimuth beam causes the enlargement in the size of the azimuth in the scene. The geometry of this phenomenon is shown in Figure 20a, where $R_{0}$ is the minimum range, $R_{max}$ is the maximum range, $\Theta_{a}$ is the array beamwidth, $\theta_{s}$ is the steering angle, and $\Theta_{as}$ is the beamwidth at $\theta_{s}$.

As demonstrated in Figure 20a, the measured scene and its size are exhibited. The array beamwidth can be calculated as follows:(28)$$\Theta_{a}\approx 0.886\frac{\lambda }{D}\left(rad\right)\approx 51\frac{\lambda }{D}\left(deg\right)$$
where λ is the wavelength and *D* is the array diameter. For an array with *N* antennas and a distance between them of λ/2, the array beamwidth becomes:(29)$$\Theta_{a}\approx 51\frac{2}{N-1}\left(deg\right).$$

If the beamwidth of the receive array is computed by N=8,
(30)$$\Theta_{a}\approx 51\frac{2}{7}\approx 14.5^{\circ }.$$

On the other hand, the array beamwidth broadens due to the steering, so the beamwidth related to the steering angle can be defined as:(31)$$\Theta_{as}=\frac{\Theta_{a}}{\cos\theta_{s}}$$
where $\theta_{0}$ represents the broadside beamwidth, θ represents the beamwidth at the angle $\theta_{s}$, and $\theta_{s}$ represents the steering angle.
(32)$$\begin{array}{cc} \Delta L & =\frac{L}{2}-R_{0}\left(\tan\left(\frac{\theta_{s}-\Theta_{as}}{2}\right)\right)\\ \; & =R_{0}\left(\tan\left(\frac{\theta_{s}+\Theta_{as}}{2}\right)-\tan\left(\frac{\theta_{s}-\Theta_{as}}{2}\right)\right)\\ \; & =R_{0}\left(\frac{\sin(2\Theta_{as})}{\cos{\left(\theta_{s}\right)}^{2}+\cos{\left(\Theta_{as}\right)}^{2}-1}\right) \end{array}$$

ΔL can be calculated via Equation (Equation 32); the corresponding scene size can then be found. The situation here is whether the beamwidth and, thus, the size of the illuminated scene vary, which scene size will be used for image formation, and what the resulting image will look like. To achieve this, the user must choose between two options. If the broadside beam is used as an example, which is $14.5^{\circ }$, the scene size for a range of 4 m will be 0.9 m × 1 m. On the other hand, if steering is applied from $-45^{\circ }$ to $45^{\circ }$, the beamwidth becomes $20.5^{\circ }$ and the result is 0.9 m × 1.45 m. Although the difference is not significant, defining the scene size in terms of broadside beamwidth simplifies the application of the concept. However, it should be remembered that the applied limit may result in overlapping or pixel loss in areas corresponding to the scene’s borders.

## 6. Conclusions

This study was prepared with the aim of creating an alternative imaging concept using recent technologies in the field of SAR imaging. The Strip-Spot concept, which allows one to obtain high-resolution images with a contiguous swath using a radar with a MIMO array antenna at high frequencies, was introduced. First of all, its geometry and signal model were presented. This concept was tested in experiments on calibrated targets at 24 GHz and on real targets at 77 GHz. According to the results of the experiments, the Strip-Spot concept enables a two-fold improvement in resolution over Stripmap mode. A performance analysis of the system was also conducted, and it was seen that, when this concept is evaluated with sample parameters, it can allow viewing at up to 190 metres for −23 dB. This is a comprehensive concept from the theoretical framework, then the simulation and experiment, to the performance analysis. On the practical side, the motion error compensation is open to investigation. It can benefit from other fields who need a higher resolution for their purposes such as detection quality for autonomous cars. This concept can be adapted to short-range GMTI and InSAR applications, resulting in a multi-functional SAR system.

## Figures and Tables

**Figure 1 sensors-22-09153-f001:**
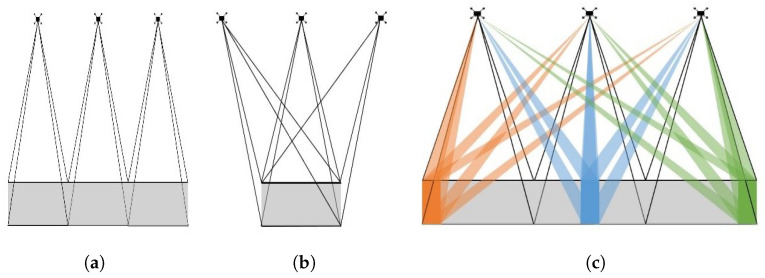
Illustration of SAR modes: (**a**) Stripmap, (**b**) Spotlight, and (**c**) Strip-Spot.

**Figure 2 sensors-22-09153-f002:**
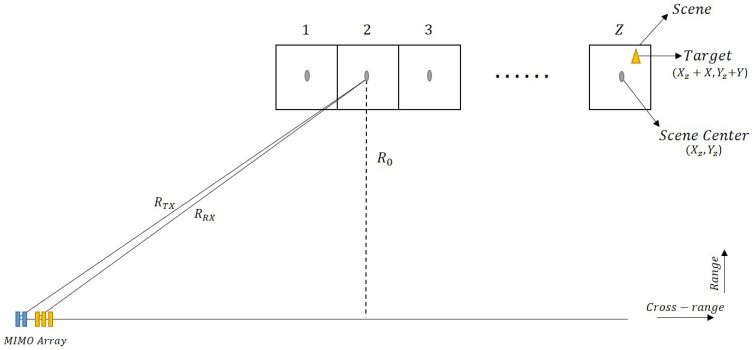
Illustration of Strip-Spot SAR geometric model.

**Figure 3 sensors-22-09153-f003:**
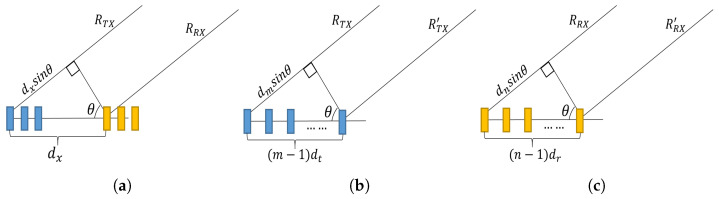
Illustration of instantaneous slant ranges from transmitter to target ($R_{Tx}$) and from target to receiver ($R_{Rx}$) (**a**) MIMO Array (**b**) Transmit Array (**c**) Receive Array.

**Figure 4 sensors-22-09153-f004:**
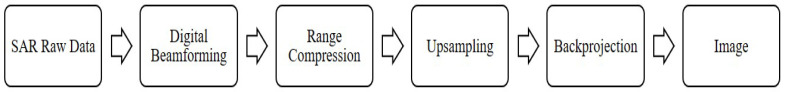
Image formation diagram for the model.

**Figure 5 sensors-22-09153-f005:**
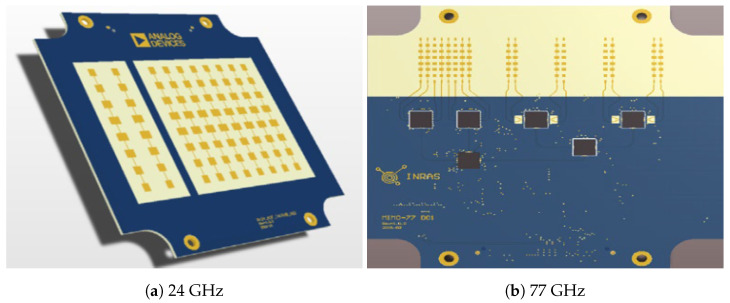
The front-ends used in our experiments.

**Figure 6 sensors-22-09153-f006:**
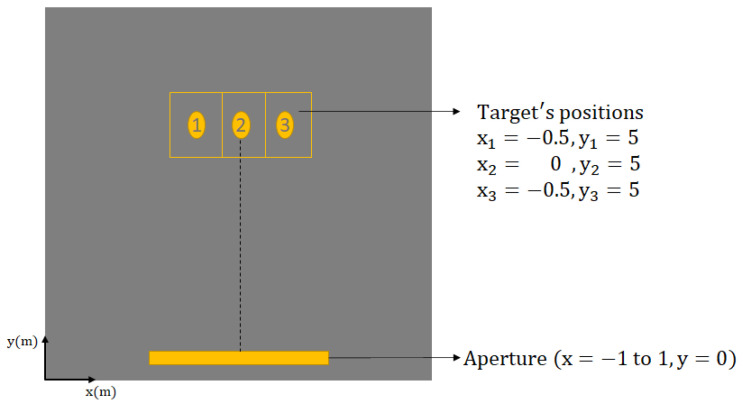
Experimental design of the 24 GHz MIMO-SAR trial.

**Figure 7 sensors-22-09153-f007:**
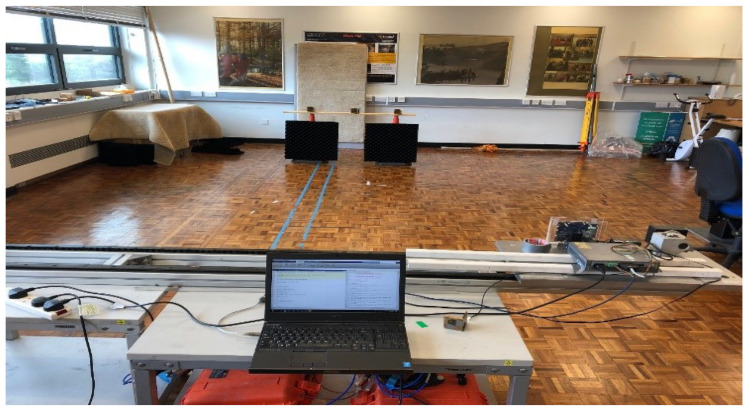
Experimental setup for the 24 GHz MIMO-SAR trial.

**Figure 8 sensors-22-09153-f008:**
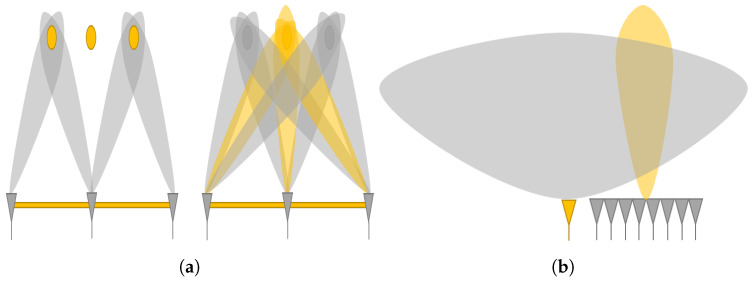
(**a**) Comparison of Spotlight and Strip-Spot modes used in the experiment; (**b**) emulated Stripmap acquisition.

**Figure 9 sensors-22-09153-f009:**
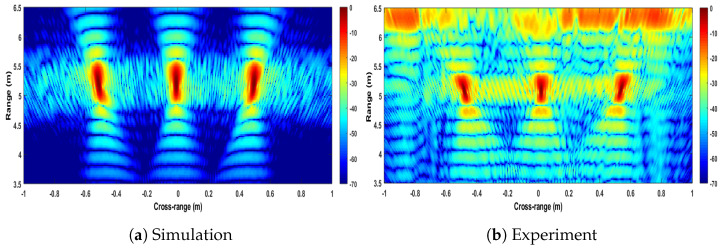
Images of Strip-Spot SAR.

**Figure 10 sensors-22-09153-f010:**
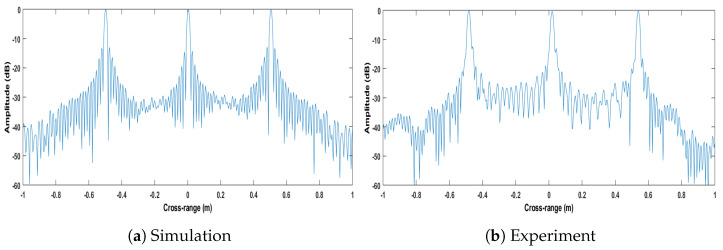
Cross-section of Strip-Spot SAR images in the cross-range dimension.

**Figure 11 sensors-22-09153-f011:**
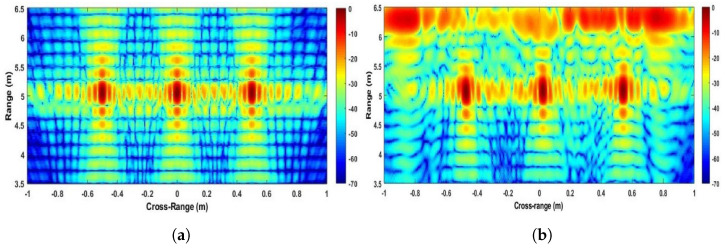
Image of Stripmap SAR for (**a**) simulation and (**b**) experiment.

**Figure 12 sensors-22-09153-f012:**
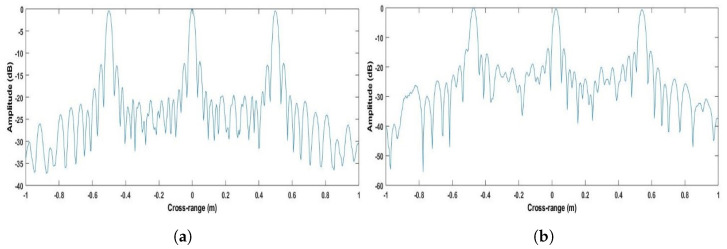
Cross-section of Stripmap SAR in the cross-range dimension for (**a**) simulation and (**b**) experiment.

**Figure 13 sensors-22-09153-f013:**
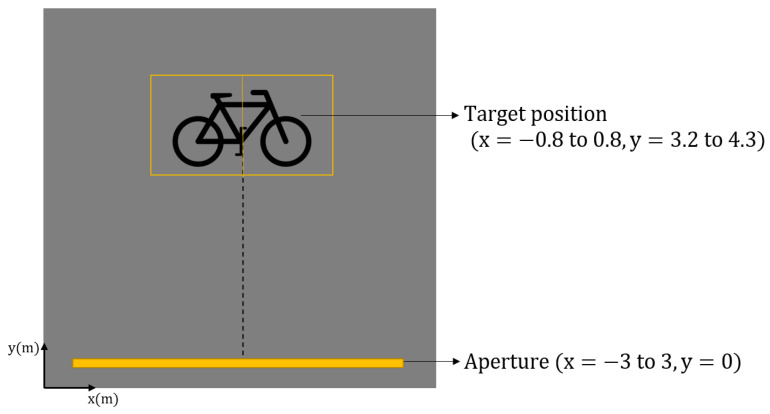
Illustration of the experiment using a bike as the target.

**Figure 14 sensors-22-09153-f014:**
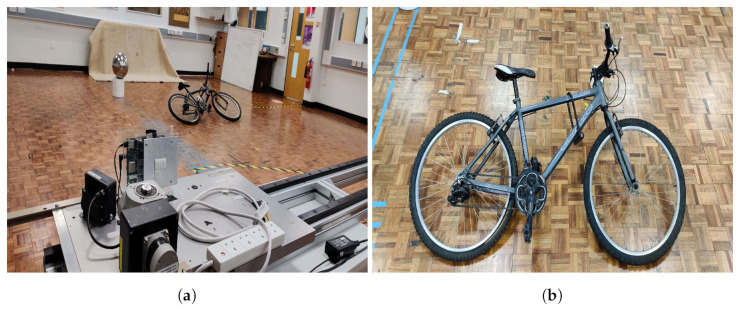
The experimental setup of the extended experiment; (**a**) equipment and (**b**) target scene.

**Figure 15 sensors-22-09153-f015:**
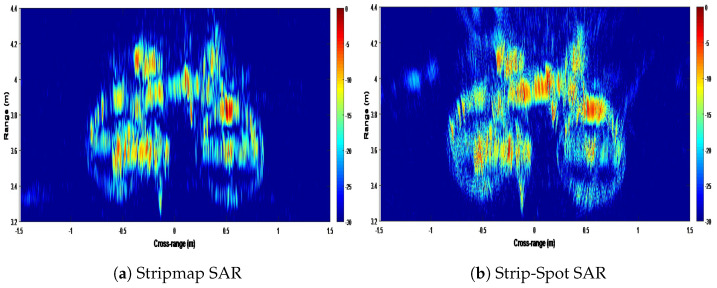
The images of the target bike.

**Figure 16 sensors-22-09153-f016:**
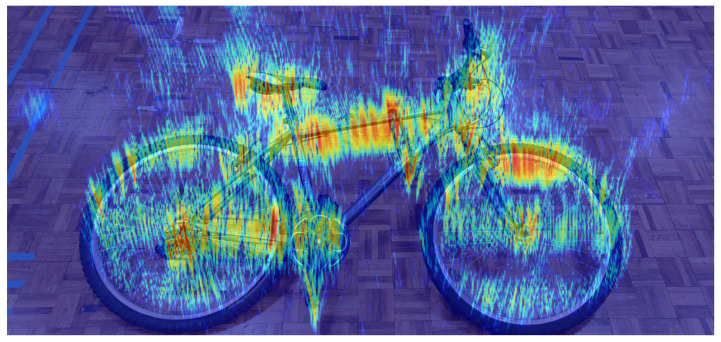
Comparison of the real bike and the result of overlapping the photo and the image.

**Figure 17 sensors-22-09153-f017:**
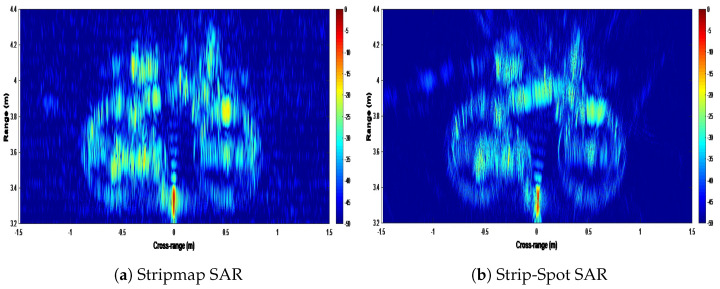
The images of the target bike with the corner reflector.

**Figure 18 sensors-22-09153-f018:**
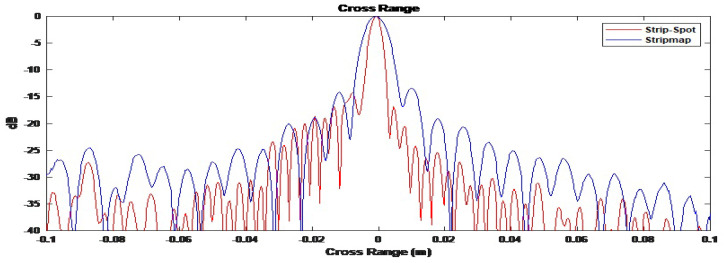
Cross-section of the cross-range of the corner reflector.

**Figure 19 sensors-22-09153-f019:**
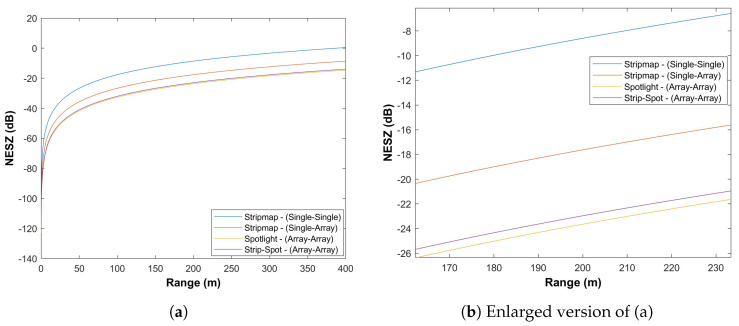
Sensitivity analysis comparison based on Table 3.

**Figure 20 sensors-22-09153-f020:**
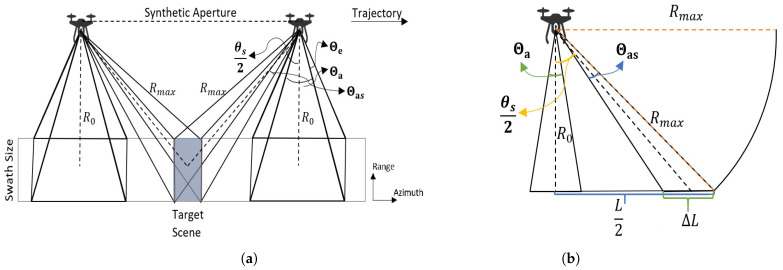
(**a**) The Strip-Spot SAR geometry, including beamwidths and steering angle. (**b**) The cross-section of the Strip-Spot acquisition to determine scene length in azimuth due to steering.

**Table 1 sensors-22-09153-t001:** The details of the INRAS radar for the front-ends at 24 GHz and 77 GHz.

Parameter	Value	Value	Unit
Type	24 GHz Front End	77 GHz Front End	
Frequency	24	77	GHz
Beamwidth-V (Tx)	12.8	13.2	Degree
Beamwidth-V (Rx)	12.8	12.8	Degree
Beamwidth-H (Tx)	76.5	51	Degree
Beamwidth-H (Rx)	76.5	76.5	Degree
Amount of Transmitter	2	4	-
Amount of Transmitter	8	8	-
Transmitter Power	8	10	dBm
Antenna Gain (Tx)	13.2	17	dBi
Antenna Gain (Rx)	13.2	15	dBi

**Table 2 sensors-22-09153-t002:** The comparison of experimental outcomes.

Mode	Frequency	Range	Aperture	Cross-Range Resolution
	**(GHz)**	**(m)**	**(m)**	**(mm)**
Stripmap	24	5.1	2	29.0
Strip-Spot	24	5.1	2	15.8
Stripmap	77	3.2	6	6.97
Strip-Spot	77	3.2	6	3.53

**Table 3 sensors-22-09153-t003:** The sensitivity formulas for different antennas and SAR modes.

Tx	Rx	Mode	Sensitivity
0.25 cm Single	Single	Stripmap	$\sigma^{0}=\frac{2V{\left(4\pi \right)}^{3}R^{3}kTFL_{s}}{P_{av}G_{Tx}G_{Rx}\lambda^{3}d_{r}}$
0.25 cm Single	Array	Stripmap	$\sigma^{0}=\frac{2V{\left(4\pi \right)}^{3}R^{3}kTFL_{s}}{P_{av}G_{Tx}G_{Rx}\lambda^{3}d_{r}}\frac{1}{N}$
0.25 cm Array	Array	Spotlight	$\sigma^{0}=\frac{2V{\left(4\pi \right)}^{3}R^{3}kTFL_{s}}{P_{av}G_{Tx}G_{Rx}\lambda^{3}d_{r}}\frac{1}{MN}$
0.25 cm Array	Array	Strip-Spot	$\sigma^{0}=\frac{2V{\left(4\pi \right)}^{3}R^{3}kTFL_{s}}{P_{av}G_{Tx}G_{Rx}\lambda^{3}d_{r}}\frac{1}{MNAF_{av}}$

**Table 4 sensors-22-09153-t004:** The parameters of sensitivity analysis.

Parameter	Symbol	Value	Unit
Transmit Power	$P_{t}$	10	dBm
Transmit Antenna Gain	$G_{Tx}$	17	dBi
Receiver Antenna Gain	$G_{Rx}$	15	dBi
Transmit Number	*M*	4	-
Receiver Number	*N*	8	-
Wavelength	λ	0.0039	m
Range resolution	$d_{r}$	0.25	m
Velocity	*V*	10	m/s
Boltzmann Constant	*k*	1.38 × $10^{-23}$	-
Temperature	*T*	295	K
Noise Figure	*F*	10	dB
Signal Loss	$L_{s}$	4	dB

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
