# Peer review of "A New Concept of Contiguous-Swath SAR Imaging with High Resolution: Strip-Spot SARâ€"

_sensors, 2022, doi:10.3390/s22239153_

Round 1
Reviewer 1 Report
The authors provide an experimental analysis of their DBF-based SAR image hybrid processing method in a lab setting. This processing technique has been thoroughly researched in satellite-based SAR, and the authors' solution is good. The following recommendations are asked to be clarified by the writers or added to the original text in order to more easily enable the readers to understand the techniques of the article.
(1) Describe the pertinent Equation 21 parameters.
(2) How should practical application handle a change in oblique angle of view?
(3) Is there a way to correct the distance migration impact caused by the cluster radar's stripe processing, which may cause the image to appear to have a major edge transformation phenomenon?
Author Response
Dear Reviewer,
We appreciate your comments. Your insightful comments contribute to the paper's improvement. We reviewed all the comments and made the necessary corrections. Also, the language was reviewed for readability before uploading the final version.
You may find your comments, as well as our responses and revisions, in the form below.
We welcome any further feedback you may have.
Regards

Reviewer 2 Report
This paper proposed a Synthetic Aperture Radar (SAR) imaging concept called Strip-Spot SAR. The effectiveness of the proposed method is demonstrated by experimental results. It can be accepted after revising the following concerns:
1. Avoid lumping references as in [x-y]. It is not necessary to give several references that say exactly the same.
2. Avoid using first person.
3. The novelty and motivation of this work is not very clear. Please rewrite the introduction to show the motivations and contributions of this work.
4. This paper compares with the "strip map" method. Is there any other literature that has studied the relevant content? Besides, are there any verification results in the actual application scenario? The comparison results or the discussion should be added.
5. The subgraphs (a) and (b) in Figure 1 are visually indistinguishable.
6. What is "Experience CRR"? Readers in this field may understand it, but the full name of the abbreviation should be given when it first appears.
7. The authors should make sure the conclusions reflect on the strengths and weaknesses of their work, how others in the field can benefit from it and thoroughly discus future work.
8. The writing of the paper needs to be further polished to make it publishable. In addition, the format of references is not uniform.
Author Response

(The authors gave the same response as above.)

Round 2
Reviewer 2 Report
It seems that the authors have addressed the requests of the auditors.